# Discrepancy between Tumor Size Assessed by Full-Field Digital Mammography or Ultrasonography (cT) and Pathology (pT) in a Multicenter Series of Breast Metaplastic Carcinoma Patients

**DOI:** 10.3390/cancers16010188

**Published:** 2023-12-30

**Authors:** Mirosława Püsküllüoğlu, Katarzyna Świderska, Aleksandra Konieczna, Wojciech Rudnicki, Renata Pacholczak-Madej, Michał Kunkiel, Aleksandra Grela-Wojewoda, Anna Mucha-Małecka, Jerzy W. Mituś, Ewa Stobiecka, Janusz Ryś, Michał Jarząb, Marek Ziobro

**Affiliations:** 1Department of Clinical Oncology, Maria Sklodowska-Curie National Research Institute of Oncology, Kraków Branch, 31-115 Kraków, Poland; aleksandra.grela-wojewoda@onkologia.krakow.pl (A.G.-W.); marek.ziobro@onkologia.krakow.pl (M.Z.); 2Breast Cancer Unit, Maria Skłodowska-Curie National Research Institute of Oncology, Gliwice Branch, 44-102 Gliwice, Poland; 3Department of Breast Cancer and Reconstructive Surgery, Maria Sklodowska-Curie National Research Institute of Oncology in Warsaw, 02-781 Warsaw, Poland; 4Department of Electroradiology, Jagiellonian University Medical College, 31-008 Kraków, Poland; wrudnicki@cmuj.pl; 5Department of Anatomy, Jagiellonian University Medical College, 31-008p Kraków, Poland; renata.pacholczak@uj.edu.pl (R.P.-M.);; 6Department of Gynaecological Oncology, Maria Sklodowska-Curie National Research Institute of Oncology, Kraków Branch, 31-115 Kraków, Poland; 7Department of Chemotherapy, The District Hospital, 34-200 Sucha Beskidzka, Poland; 8Department of Radiotherapy, Maria Sklodowska-Curie National Research Institute of Oncology, Kraków Branch, 31-115 Kraków, Poland; 9Department of Surgical Oncology, Maria Sklodowska-Curie National Research Institute of Oncology, Kraków Branch, 31-115 Kraków, Poland; 10Department of Pathology, Maria Skłodowska-Curie National Research Institute of Oncology, Gliwice Branch, 44-102 Gliwice, Poland; 11Department of Tumour Pathology, Maria Sklodowska-Curie National Research Institute of Oncology, Kraków Branch, 31-115 Kraków, Poland

**Keywords:** metaplastic breast cancer, tumor size assessment, full-field digital mammography, ultrasonography

## Abstract

**Simple Summary:**

In this study on metaplastic breast cancer (BC-Mp), we assessed the accuracy of predicting tumor size measurements in pathology examinations (pT) using (1) preoperative breast ultrasound and (2) full-field digital mammography (cT). The retrospective multi-center cohort study included 45 females with BC-Mp who underwent upfront surgery. Results showed that tumor sizes were statistically assessed as significantly smaller compared to the pathological examination in both imaging methods. Neither of the techniques showed clear statistical superiority, but both were more accurate for smaller tumors. The risk of underestimating tumor size with these methods, especially in larger tumors, needs consideration in surgical planning for BC-Mp. Neither ultrasound nor full-field digital mammography can be recommended as superior in this diagnostic setting. Tumor size was found to be the sole predictive factor for BC-Mp, emphasizing the crucial role of its assessment.

**Abstract:**

Metaplastic breast cancer (BC-Mp) presents diagnostic and therapeutic complexities, with scant literature available. Correct assessment of tumor size by ultrasound (US) and full-field digital mammography (FFDM) is crucial for treatment planning. Methods: A retrospective cohort study was conducted on databases encompassing records of BC patients (2012–2022) at the National Research Institutes of Oncology (Warsaw, Gliwice and Krakow Branches). Inclusion criteria comprised confirmed diagnosis in postsurgical pathology reports with tumor size details (pT) and availability of tumor size from preoperative US and/or FFDM. Patients subjected to neoadjuvant systemic treatment were excluded. Demographics and clinicopathological data were gathered. Results: Forty-five females were included. A total of 86.7% were triple-negative. The median age was 66 years (range: 33–89). The median pT was 41.63 mm (6–130), and eight patients were N-positive. Median tumor size assessed by US and FFDM was 31.81 mm (9–100) and 34.14 mm (0–120), respectively. Neither technique demonstrated superiority (*p* > 0.05), but they both underestimated the tumor size (*p* = 0.002 for US and *p* = 0.018 for FFDM). Smaller tumors (pT1-2) were statistically more accurately assessed by any technique (*p* < 0.001). Only pT correlated with overall survival. Conclusion: The risk of underestimation in tumor size assessment with US and FFDM has to be taken into consideration while planning surgical procedures for BC-Mp.

## 1. Introduction

Breast cancer (BC) continues to be the leading cause of mortality among women globally [1]. It is categorized into various subtypes: Luminal A and B, Human epidermal growth factor receptor 2 (HER-2) positive, and basal-like (with triple-negative being the most prevalent among this subtype), based on the expression of estrogen receptor (ER), progesterone receptor (PR), HER-2 and Ki-67 status [2].

The majority of breast malignancies originate from epithelial components [3]. Ductal carcinoma (not-otherwise specified [NOS]) accounts for the majority of all cases, while lobular carcinoma represents 5–15% of cases [3,4]. Less common subtypes, including neuroendocrine, mucinous, tubular, and medullary carcinomas, each constitute only approximately 1–2% of cases [3]. Primary metaplastic breast cancer (BC-Mp) is an extremely rare and heterogeneous histotype, comprising around 1% of all BCs [3]. Due to their diverse nature and rarity, there is ongoing controversy regarding prognostic factors and treatment guidelines. BC-Mp is typically characterized as triple negative and generally associated with a poor prognosis [5,6]. It is claimed that BC-Mp have low chemosensitivity, and responses to neoadjuvant systemic treatment remains unsatisfactory [7].

Full-field digital mammography (FFDM) serves as the primary diagnostic imaging approach for BC, encompassing clinical cases and screening scenarios. This iterative method gauges tumor size, location, calcification presence, and other malignancy indicators, with retrospective viewing possible. FFDM’s effectiveness is curtailed by dense, glandular breasts (American College of Radiology type D) [8]. Ultrasonography (US) supplements BC diagnostic imaging, excelling in dense breast scenarios compared to FFDM. Nevertheless, its specificity is modest, contingent upon the examiner’s proficiency [9].

Both FFDM and US are widely accessible imaging modalities that do not necessitate contrast agents, rendering them the preferred initial options for tumor evaluation. Consequently, FFDM and US are frequently employed for preoperative tumor size determination. While their performance is inferior to contrast-enhanced techniques like magnetic resonance imaging (MRI) or contrast-enhanced mammography, the marginal disparities do not overshadow the advantages of availability and ease of use, establishing FFDM and US as the primary choices for tumor size assessment [10,11].

The existing literature on BC-Mp and diagnostic imaging methods is limited (see Table 1); however, both FFDM and US offer distinct attributes for this cancer type, justifying their role as fundamental imaging modalities for assessing tumor size [12,13].

The objective of this study was to check the accuracy of full-field digital mammography and ultrasonography in assessing the tumor size in the postoperative pathology report of patients with metaplastic breast cancer operated on at three Cancer Reference Centers in Southern and Central Poland.

## 2. Materials and Methods

### 2.1. Patients

Patients diagnosed between 2012 and 2022 with BC-Mp were identified from the registry systems of the Maria Sklodowska-Curie National Research Institute of Oncology Branch Gliwice, Warsaw, and Krakow.

The inclusion criteria for the study were a diagnosis of BC-Mp in postsurgical pathology report, and assessment of tumor size by US and/or FFDM performed in the National Cancer Centers before the operation (not earlier than 60 days before the surgery; if multiple reports were available, then the one closest to the surgery was included). The patients who received neoadjuvant systemic treatment and were without an original pathology report were excluded from the study. Patients with coexisting active malignancies were also excluded. There were no restrictions regarding the sex or age of the patients.

Data regarding sex and age; clinical data regarding tumor location, clinical staging, dates and types of treatment, size of the tumor in ultrasound and mammography; and histopathological data (including histology, status of ER, PR, HER-2, Ki-67, presence of ductal carcinoma in situ (DCIS), tumor grade, and presence of different BC-Mp components) were gathered retrospectively.

The size of the tumor assessed clinically (cT) is the largest dimension visible in a specific examination. In US, the transducer should be rotated until obtaining a cross-section with the largest dimension. In FFDM, cranio-caudal (CC) or medio-lateral oblique (MLO) projection is selected based on the location of the largest tumor dimension. To determine the second dimension, measurements are made perpendicular to the axis of the largest tumor dimension.

The diagnosis of BC-Mp was commonly established through a combination of morphological assessment and immunohistochemical staining [26]. During the preparation of the breast tissue specimen, a pathologist assesses the primary tumor size (pT) by measuring its largest dimension and perpendicular axes to it as per College of American Pathologists guidelines.

### 2.2. Ethical Approval

The ethical approval was obtained from the Maria Sklodowska-Curie National Research Institute of Oncology Branch Krakow Ethical Committee, as confirmed by decision number 3/2023. Considering the retrospective design of the study, written informed consent was not sought from the patients in accordance with the decision made by the Ethical Committee.

### 2.3. Statistical Analysis

R version 4.3.1. was used for computations (https://www.r-project.org, accessed on 10 August 2023). Distributions of quantitative variables were summarized with mean, standard deviation, median and quartiles, whereas distributions of qualitative variables were summarized with number and percent of occurrence for each of their values.

Mann–Whitney test was used to compare quantitative variables between two groups, while Kruskal–Wallis test (followed by Dunn post hoc test) was used for more than two groups. Relationship between two quantitative variables was assessed with Spearman’s coefficient of correlation. Paired Wilcoxon test was used to compare two repeated measures of quantitative variables. Cox proportional hazards model was used to analyze the impact of potential predictors on a survival times. Hazard rations (HR) with 95% confidence intervals were shown. Significance level for all statistical tests was set to 0.05.

## 3. Results

### 3.1. Characteristic of the Population

This study included 45 female patients with no male patients, and all of them had unilateral tumors. The median age at diagnosis was 66 years, spanning from 33 to 89 years, while the mean age was 64.2 years. BC-Mp accounted for less than 1% of the breast cancer patients in the Maria Sklodowska-Curie National Research Institute of Oncology hospital registry systems. Figure 1 and Figure 2 presents exemplary BC-Mp pictures in FFDM and US.

Regarding the BC subtype, six patients were Luminal B (including five PR negative), none were HER-2 positive, and thirty-nine were triple-negative. The median Ki67 was 45% (range: 5–95%). The median tumor size at histopathology diagnosis was 30 mm (range: 6–130 mm), and the mean was 41.6mm. The most common differentiation found was squamous in 14 (31.1%) cases. Until June 2023, fifteen patients died, six experienced local recurrence, and seven distal metastases. Further pathological, diagnostic and clinical data regarding patients are presented in Table 2.

### 3.2. In Both Imaging Techniques, Tumor Size Was Found to Be Statistically Smaller Than in Pathomorphology

In both imaging methods, the tumor size was statistically assessed as significantly smaller compared to the pathomorphological examination (Table 3). Figure 3 presents a comparison of tumor size assessments for each patient.

When the difference in tumor size between US and pathomorphological examination or FFDM and pathomorphological examination was compared, no statistical significance was seen (mean ± SD 10.55 ± 14.98 and 10.55 ± 14.98 respectively, *p* = 0.867) (see Figure 3c), showing no clear advantage of any imaging technique.

### 3.3. Tumor Size’s Sole Significance in Accurate Imaging Assessment

Tumor size assessed in the pathology report (pT) is the sole factor that displayed a statistically significant impact on the accuracy of tumor assessment through both imaging techniques (Figure 4). The correlation is positive (Spearman’s correlation coefficient: 0.516 and 0.541 for FFDM and US respectively, *p* < 0.001); thus, as the size of the tumor (pT) increases, there is an augmented disparity observed between imaging findings (cT) and the pathological examination (pT). The incongruity between imaging findings and pathological examination was significantly greater in tumors classified as pT3-4 than pT1-2 (tumors ≤ 50 mm) (strong significance with *p* < 0.001 in the Mann–Whitney test).

Conversely, variables including patients’ age, method of initial diagnosis, body mass index (BMI), menopausal status, lymph nodes involvement, tumor grading, Ki67 or ER/PR status did not demonstrate a statistically significant effect on the precision of tumor size evaluation using these imaging techniques.

### 3.4. Lymph Nodes Involvement (pN) Does Not Correlate with Tumor Size (pT)

Tumor size does not statistically correlate with lymph nodes involvement in this patient cohort (*p* = 0.133 from the Mann–Whitney test) (Figure 5).

### 3.5. Tumor Size (pT) Correlates with Treatment Outcomes

The Univariate Cox proportional hazards models revealed that pT is found to correlate with overall survival (OS) and disease-free survival (DFS). Each additional millimeter of the tumor increases the probability of mortality at any given time by 3.5% (HR = 1.035; CI 1.011–1.041, *p* = 0.001). Furthermore, each additional millimeter of the tumor raises the probability of recurrence, metastasis, or mortality at any given time by 2.6% (HR = 1.026, CI 1.018–1.052, *p* < 0.001).

Each subsequent patient’s year of life increases the probability of mortality at any given time by 4.2% (hazard ratio = 1.042, 95% confidence interval 1.002–1.083, *p* = 0.041). However, this observation did not attain statistical significance in terms of disease-free survival (DFS). Other factors such as lymph node status, consistency between pre-and postoperative histopathology (BC-Mp diagnosis), presence of different components (NST, DCIS, squamous), tumor grading, hormonal status, BMI, the method the tumor was detected, and menopausal status did not reach statistical significance in terms of influence on DFS or OS.

## 4. Discussion

There is limited data available concerning preoperative tumor assessment (cT) in patients with BC-Mp (Table 1). We have successfully compiled one of the most extensive patient multi-center cohorts with real-world data pertaining to tumor size in imaging studies. Only the study conducted by Langlands and colleagues included a larger number of patients with BC-Mp with data from their imaging studies. However, the authors there did not endeavor to address the question of which examination could more accurately predict tumor size as assessed through pathological examination. In this context, our study stands as the largest in terms of the number of patients [23].

Available studies highlight that findings from US can suggest BC-Mp presence. US showed BC-Mp with complex echogenicity, solid–cystic components, circumscribed margins, and posterior acoustic enhancement, differentiating it from other subtypes. Increased color flow signals and high resistance indices in feeding arteries were also seen [12,13,16,19,21]. FFDM aided in detection and assessment but did not offer highly distinctive features of BC-Mp. FFDM typically depicted BC-Mp as circumscribed, noncalcified masses with both circumscribed and spiculated portions, similar to other subtypes, lacking unique distinguishing traits [16,17,19]. The diagnosis of BC-Mp still relies primarily on histopathological results. Furthermore, some studies suggest that no definitive characteristics can be attributed to any of the imaging techniques [19].

Certain data on BC-Mp suggest that these tumors are unresponsive to systemic preoperative treatment [7]. Thus, when technically feasible, initial surgery can be considered for these patients. In this context, accurate cT assessment, preferably correlated with post-operative pT assessment, is crucial for planning the extent and type of surgical intervention. However, other researches claim that although responses to the neoadjuvant treatment are low, they do not vary from the general BC population [27]. Moreover, the majority of BC-Mp tumors are TNBC. As new treatment options are currently available in early TNBC, including check-point inhibitors for planning the proper treatment pathway for BC-Mp patients is still an open issue [28,29,30].

There is no consensus regarding which factors influence the prognosis of BC-Mp patients. The study conducted by Oberman and colleagues concentrated on the correlation between neoplasm size and patient prognosis, a correlation that our study corroborated. In their study, tumors smaller than 4 cm exhibited a more favorable course. Furthermore, it is noteworthy that Oberman and colleagues did not investigate whether lymph node status correlated with prognosis, as they only identified one patient with lymph node involvement, but similarly to our group, they also described the lack of correlation of microscopic findings with prognosis [18].

More data regarding the correlation between tumor size in US/FFDM and pT are available for typical histological subtypes of BC (NOS or lobular), as well as the entire BC population. FFDM and US are widely employed to estimate tumor size before initiating treatment. Both methods are statistically reliable predictors of histopathological tumor size. However, FFDM tends to overestimate, while US generally underestimates tumor size in most cases of BC [31,32,33]. Different trends were seen in our cohort of patients with BC-Mp as both imaging methods underestimated the size of the primary tumor. The reason for that can be the already-mentioned distinctive morphology of BC-Mp that facilitates more straightforward dimensioning, primarily attributable to well-defined lesion margins. In contrast, NOS or lobular cancers frequently exhibit irregular or indistinct boundaries. The most accurate method for measuring breast tumor size is MRI. However, it is less available and not routinely performed in patients diagnosed with BC-Mp, primarily due to the common unifocal nature of the disease. Additionally, MRI tends to overestimate tumor size [33].

One of the certain limitations in the study is the retrospective nature of the study. Due to the low incidence of this malignancy and the fact that a large percentage of the patients are diagnosed with BC-Mp only when the whole tumor is assessed (postoperatively), it would be challenging to performed prospective observation. Additionally, we have not collected other data from FFDM and US, such as tumor morphology, presence of calcifications or additional features in surrounding tissue. Images were also not reassessed by the breast radiologist. Clinicians rarely have the ability to assess and incorporate these radiological findings into their treatment decisions, and furthermore, these additional findings are not included in the diagnostic or treatment guidelines.

Another limitation is the size of the population. However, our cohort of patients remains the second largest published regarding BC-Mp patients, and currently available reports lack data regarding size of the tumor in imaging studies.

Metaplastic breast cancer (BC-Mp) is not only rare, but is also characterized by histological and immunohistochemical heterogeneity. There is a need for studies with larger cohorts to explore different clinical behaviors among various subtypes of BC-Mp. Currently, BC-Mp is often considered to respond poorly to treatment [7]. However, other studies indicate higher HER2-positive status than in our cohort, increased programmed cell death ligand-1 (PDL1) positivity, and the presence of specific somatic tumor mutations and germline mutations like *Breast Cancer susceptibility gen1* mutation [34,35,36,37,38]. This suggests the potential for targeted and possibly more effective systemic treatment, including neoadjuvant systemic approaches. As treatment strategies evolve, it will be essential to validate imaging modalities for assessing treatment responses in order to guide appropriate surgical management.

## 5. Conclusions

Neither of these modalities, US or FFDM, demonstrates superiority in accurately assessing the size of the tumor mass, as indicated in the pathology report (pT). However, they both underestimate the size with statistical significance. While BC-Mp pT1-2 primary tumors undergo more precise evaluation, it is essential to recognize that larger masses are associated with an increased likelihood of size disparity between FFDM or US findings (cT) and actual pathology dimensions (pT). Prospective studies with properly trained radiologists should be planned for this population to obtain better knowledge about more effective diagnostic strategies or better applications of existing ones.

## Figures and Tables

**Figure 1 cancers-16-00188-f001:**
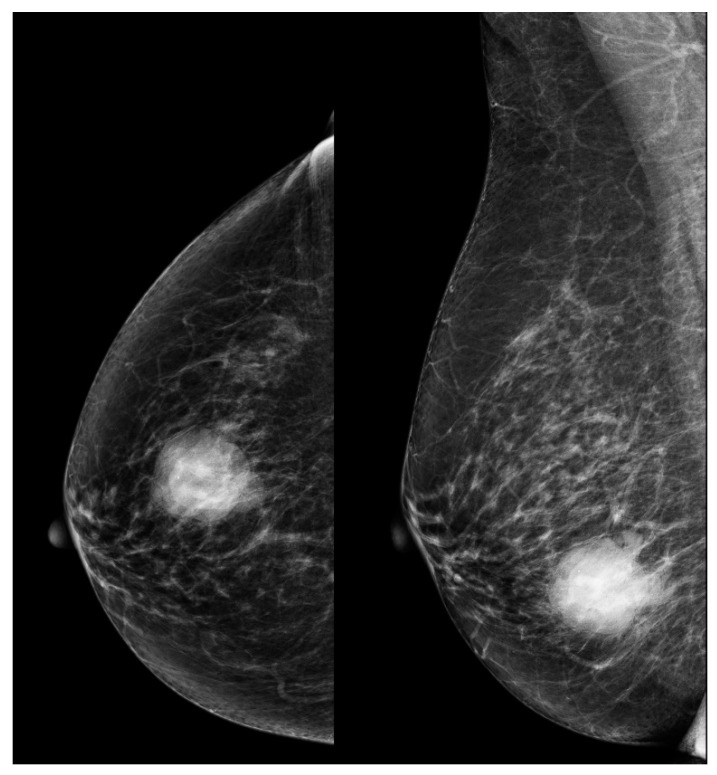
Full-field digital mammography images of metaplastic breast cancer: a lesion at 6 o’clock in the middle 1/3 of the right breast with well-defined edges, lobulated shape with high density. Right craniocaudal view on the left and right mediolateral oblique view on the right. (Images courtesy of the Diagnostic Imaging Department, University Hospital in Krakow.).

**Figure 2 cancers-16-00188-f002:**
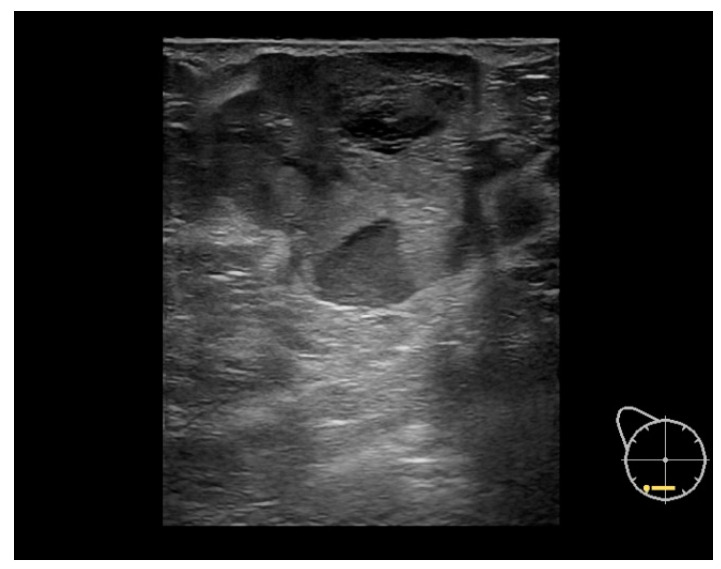
Ultrasound image of metaplastic breast cancer. A hypoechogenic lesion at 6 o’clock in the right breast with well-defined edges, lobulated shape with necrosis areas. (Image courtesy of the Diagnostic Imaging Department, University Hospital in Krakow.).

**Figure 3 cancers-16-00188-f003:**
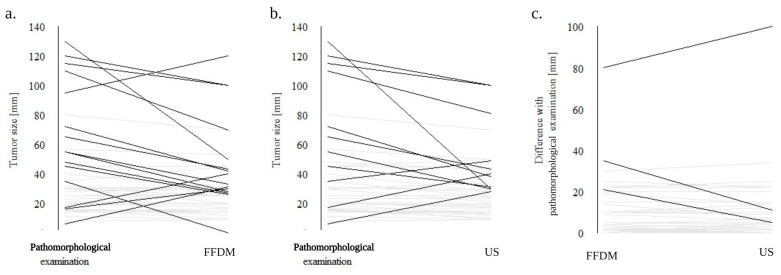
Primary tumor size (T) assessed by different techniques: (**a**). Comparison between pathomorphological assessment (pT) and FFDM (cT). (**b**). Comparison between pathomorphological assessment (pT) and US (cT). (**c**). Comparison between FFDM and US. Each line represents different patient, patients with similar outcomes as seen as gray, patients with larger differences in tumor size assessment are presented as black lines. Abbreviations: FFDM, full-field digital mammography; US, ultrasound.

**Figure 4 cancers-16-00188-f004:**
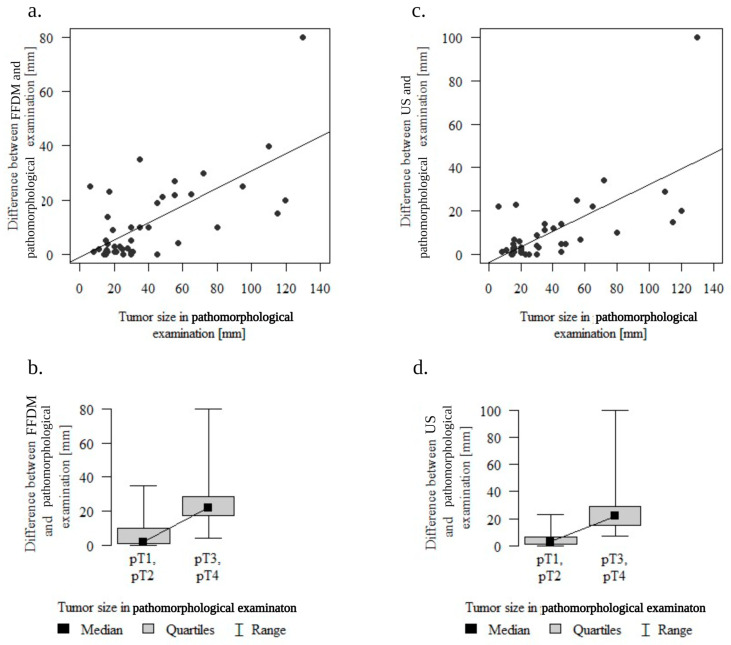
Tumor size as the factor influencing the accuracy of tumor assessment by imaging techniques. (**a**). The correlation between FFDM findings and pathological examination tumor size difference and the tumor size (**b**). The correlation between FFDM findings and pathological examination tumor size difference and the tumor size (pT1-2 vs. pT3-4) (**c**). The correlation between US findings and pathological examination tumor size difference and the tumor size (**d**). The correlation between US findings and pathological examination tumor size difference and the tumor size (pT1-2 vs. pT3-4). Abbreviations: FFDM—full-field digital mammography; US—ultrasound.

**Figure 5 cancers-16-00188-f005:**
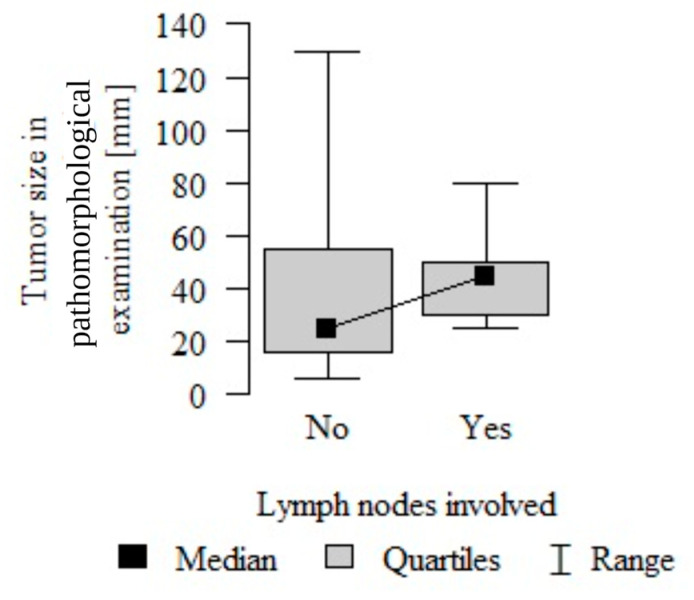
Correlation between lymph node involvement (pN) and tumor size (pT).

**Table 1 cancers-16-00188-t001:** Literature data regarding imaging studies in metaplastic breast cancer [12,13,14,15,16,17,18,19,20,21,22,23].

Study	*n*	IHC	Histopathology	Tumor Size, Median (Range)	Treatment	Conclusions
[16]	19	ER neg 91% PR neg 81% HER2 neg 80.0%	Axillary lymph node metastases *n* = 4 -Squamous cell carcinoma *n* = 10 -Mesenchymal differentiation with chondroid and mixed metaplasia *n* = 3 -Spindle carcinoma *n* = 3 -Low-grade adenosquamous carcinoma *n* = 1	No data	Surgery *n* = 19 NCT *n* = 3 Adjuvant CTH *n* = 11, RTH *n* = 3, Endocrine therapy *n* = 1	Imaging features of BC-Mp on FFDM and US were benign. T2WI MRI showed characteristic features of signal intensity using TIC curve and ADC analysis. Limitations: small sample size, lack of information regarding tumor size.
[13]	16	ER neg 83% PR neg 83% HER2 neg 43%	Axillary lymph node metastases *n* = 6 -Squamous cell type in *n* = 7 -Cartilaginous *n* = 5 -Spindle *n* = 4 -Matrix-producing *n* = 4 -Chondrosarcomatous *n* = 2 -Osseous *n* = 2	FFDM 4.6 cm (1–10) Pathologic exam. 4.2 cm (1.2–11)	Surgery *n* = 15	In US present complex echogenicity with solid and cystic components. Those findings are related to hemorrhagic or cystic necrosis found pathologically. Limitations: small sample size, no data on systemic treatment or RTH, Asiatic population, age of patients around 10 years younger than in other reports.
[17]	10	No data	Axillary lymph node metastases *n* = 1 -Squamous and spindle *n* = 3 -Spindle *n* = 6	FFDM 3 cm (1.7–6.5) Pathologic exam. 2.8 cm (1.7–9)	No data	On FFDM presents as predominately circumscribed, noncalcified masses. Characteristic feature is a circumscribed portion with a spiculated portion. Limitations: small sample, lack of data regarding subtype and treatment.
[18]	29	ER, PR neg 100%	Axillary lymph node metastases *n* = 0 -Spindle cell carcinoma *n* = 8	Pathologic exam. 5.3 cm (3.5–9)	Surgery *n* = 6 RTH *n* = 0	The paucity of lymph node metastases. The size of neoplasm correlated with prognosis. Tumors < 4 cm had a favorable coursespo The lack of correlation of microscopic findings with prognosis. Limitations: small sample size of different histopathological entities.
		ER, PR neg 100%	Axillary lymph node metastases *n* = 1 -Squamous cell metaplasia *n* = 6	Pathologic exam. 4.5 cm (1.5–6)	Surgery *n* = 6 RTH *n* = 1	
		ER, PR neg 80%	Axillary lymph node metastases *n* = 1 -Pseudosarcomatous metaplasia (spindle cell, osseous, osteoid, chondroid, rhabdomyoid elements) *n* = 15	Pathologic exam. 4.8 cm (1.5–19)	Surgery *n* = 15 RTH *n* = 1	
[12]	11	ER, PR neg 82%, HER2 neg 91%	Axillary lymph node metastases *n* = 4 -Squamous subtype *n* = 3 -Carcinosarcoma *n* = 3 -Matrix producing BC-Mp *n* = 3 -Spindle cell subtype *n* = 2	Imaging 2.7 cm (1.1–10)	No data	FFDM results of BC-Mp were similar to other subtypes. US may differentiate between BC-Mp and other subtypes due to some specific features of BC-Mp: circumscribed margins, complex echogenicity and posterior acoustic enhancement. Limitations: small sample size, retrospective study
[19]	22	ER, PR neg 95% HER2 neg 82%	Distant metastases *n* = 4 -Squamous cell *n* = 10 -Spindle cell *n* = 5 -Mesenchymal differentiation *n* = 5 -Mixed *n* = 2	US 5.4 cm (0.8–20.4) Pathologic exam. 5.9 cm	CTH = 19 Surgery *n* = 16 RTH = 13	No definite characteristic on imaging. On US and FFDM a mass with irregular shape and indistinct margins with posterior acoustic enhancement and rarely containing microcalcifications. Limitations: most of the data from the hospital electronic system were missing.
[20]	33	ER neg 91% PR neg 81% HER2 neg 84%	Axillary lymph node metastases *n* = 8 -Squamous cell *n* = 18 -Matrix producing *n* = 10 -Other heterogeneous *n* = 4 (squamous + spindle-cell type *n* = 1, spindle + osseous *n* = 1, myxoid matrix + sarcoma *n* = 1, spindle + neuroendocrine *n* = 4)	No data	Mastectomy *n* = 23 BCS *n* = 8 Excision *n* = 1	BC-Mp might display more benign features and less axillary lymph node metastasis than other BC. High signal intensity on T2 MRI images and hormone receptor negativity are most typical hallmark. Limitations: lack of data regarding tumor diameter, retrospective study
[21]	10	ER neg 90% PR neg 100% HER2 neg 70%	Axillary lymph node metastases *n* = 3 -Adenosquamous carcinoma *n* = 2 -Adenocarcinoma with spindle cell metaplasia *n* = 4 -Carcinoma with chondroid differentiation *n* = 3 -Carcinoma with osseous differentiation *n* = 1	US 5.7 cm (1–15)	Mastectomy *n* = 7 BCS *n* = 3 NCT *n* = 2 Adjuvant CTH *n* = 3	US reveals gently lobulated, complex mass lesion with cystic parts and posterior acoustic enhancement, which represents necrosis or hemorrhage. Another finding on US is increased color flow signals, and relative high RI of the feeding arteries were also seen. Limitations: small sample size, lack of data regarding tumor size in a pathological exam.
[22]	12	ER neg 100% PR neg 100% HER2 neg 83.3%	Axillary lymph node metastases *n* = 3 -Carcinoma with chondroid metaplasia *n* = 6 -Squamous large cell keratinizing *n* = 5 -Carcinosarcoma *n* = 1	Pathologic exam. 2.6 cm (1.2–4.5)	No data	On MRI, the high signal intensity on T2-weighted is useful for preoperative diagnosis of BC-Mp. It is necessary to differentiate from mucinous carcinoma and necrotic infiltrating ductal carcinoma. Limitations: small sample size, lack of data regarding treatment modalities.
[23]	71	ER neg 90% PR neg 79% HER2 neg 84.5%	Axillary lymph node metastases *n* = 11	US 2.6 cm (3–70) FFDM 3.2 cm (10–76) Pathologic exam 2.7 cm (5–75)	Mastectomy *n* = 35 BCS *n* = 36 NCT = 11	MRI outlines the extent of disease well and is useful in monitoring response to NCT. Those tumors respond by showing concentric reduction in size. Limitations: no data on histopathologic subtypes.
[14]	5	ER neg 100% PR neg 100% HER2 neg 40%	Axillary lymph node metastases *n* = 1 -Low-grade adenosquamous carcinoma *n* = 2 -Low-grade adenosquamous and mixed metaplastic carcinoma *n* = 2 -Squamous *n* = 1	No data	Mastectomy *n* = 3 BCS *n* = 1 RTH *n* = 3 Adjuvant CTH *n* = 3	BC-Mp tends to show more benign imaging features such as round or oval shape with circumscribed margins and less axillary lymph node metastasis compared with invasive ductal carcinoma. High signal intensity on T2-weighted magnetic resonance imaging due to its cystic or necrotic component may be useful for diagnosis of metaplastic carcinoma. Limitations: case series study, small sample size, lack of data regarding tumor size.
[15]	13	ER neg 86.7% PR neg 93.3% HER2 neg 93.3%	Axillary lymph node metastases *n* = 1 -Squamous cell carcinoma *n* = 8 -Spindle cell carcinoma *n* = 3 -Matrix-producing carcinoma *n* = 2 -Fibromatosis-like carcinoma *n* = 1 -Mixed metaplastic carcinoma *n* = 1	No data	No data	Imaging examinations have certain imaging features such as posterior acoustic enhancement, and together with an absence of hormone receptor expression, may suggest the metaplastic carcinoma. Limitations: lack of data regarding tumor size and treatment modalities.

Some studies described only radiologic features of BC-Mp and were not included in Table 1 [24,25]. Abbreviations: ADC; apparent diffusion coefficient, BCS; breast conserving surgery; CTH, chemotherapy; ER, estrogen receptor; exam., examination; FFDM, full-field digital mammography, HER2, human epidermal growth factor receptor; IHC, immunohistochemistry; BC-Mp, metaplastic breast cancer; MRI, magnetic resonance imaging; neg., negative; NCT, neoadjuvant chemotherapy; PR, progesterone receptor; RTH, radiotherapy; TIC, time-intensity curve; US, ultrasonography.

**Table 2 cancers-16-00188-t002:** Patients with metaplastic breast cancer characteristics.

Characteristics		*n*	*n*%
Location—side	Right breast	19	42.2
	Left breast	26	57.8
Location—quadrant	Upper outer	23	51.1
	Upper inner	1	2.2
	Lower outer	6	13.3
	Lower inner	1	2.2
	Central	4	8.9
	Not assessed/multiple	10	22.2
AJCC stage	I	16	35.6
	II	16	35.6
	III	13	28.9
pT	1	16	35.6
	2	17	37.8
	3	11	24.4
	4	1	2.2
Lymph node involvement	Negative	37	82.2
	Positive	8	17.8
Grade	I	0	0
	II	9	20.0
	III	35	77.8
	No data	1	2.2
DCIS presence	Yes	16	35.6
	No *	29	64.4
ER status	Positive	6	13.3
	Negative	39	86.7
PR status	Positive	1	2.2
	Negative	44	97.8
Subtype	Luminal A	0	0
	Luminal B	6	13.3
	HER-2 positive	0	0
	Triple-negative	39	86.7
HER2	Positive	0	0
	Negative **	45	100
Menopausal status	Postmenopausal	34	75.6
	Premenopausal	11	24.4
Method of tumor initial detection	Noted by the patient	27	60.0
	Accidental finding on imaging tests	17	37.8
	No data	1	2.2
Type of imaging technique preoperative assessment	FFDM	43	95.6
	US	41	91.1

* Or presence in hist-pat report not mentioned. ** ICH: HER2-0; HER2-1; HER2 ICH 2 and FISH negative. Abbreviations: AJCC, American Joint Committee on Cancer, 8th edition; BC-Mp, metaplastic breast cancer; ICH, immunohistochemistry; DCIS, ductal carcinoma in situ; ER, estrogen receptor; FFDM, full-field digital mammography; HER2, human epidermal growth factor receptor 2; *n*, regional lymph nodes; PR, progesterone receptor; pT, tumor; US, ultrasound.

**Table 3 cancers-16-00188-t003:** Primary tumor size (T) assessed by different techniques.

Assessment Type	Tumor Size (mm)			*p*
	Mean (±SD)	Median	Quartiles	
Pathologic examination *	40.55 ± 32.22	30	16.5–51.5	*p* = 0.018 ***
FFDM *	34.14 ± 25.19	28	18–42.5	
Pathological examination **	38.67 ± 31.84	30	16–45	*p* = 0.002 ***
US **	31.8 ± 22.38	28	17–40	

* Data for patients with available FFDM tumor size. ** Data for patients with available US tumor size. *** Wilcoxon paired test. Abbreviations: FFDM—full-field digital mammography; US—ultrasound.

## Data Availability

Data available on request.

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
