# Peer review of "Discrepancy between Tumor Size Assessed by Full-Field Digital Mammography or Ultrasonography (cT) and Pathology (pT) in a Multicenter Series of Breast Metaplastic Carcinoma Patients"

_cancers, 2023, doi:10.3390/cancers16010188_

Round 1

Reviewer 1 Report

Comments and Suggestions for Authors

The authors assess the accuracy of predicting tumor size measurements in pathology examinations using preoperative breast ultrasound and full-field digital mammography with retrospective multi-center cohort of 45 females with metaplastic breast cancers (BC-Mp). The results have demonstrated the importance of the tumor size for the evaluation of BC-Mp. And it is valuable for evaluating the tumor size and its type before choosing suitable therapy for the breast cancer. There are a few suggestions for the authors.

1. There are no more details about how the tumor size is measured with the ultrasound and FFDM. Both of them are two dimensional and it will definitely introduce different errors when imaging in different views. Whether this bring systematic errors for the tumor size evaluation is unclear. Meanwhile, how the tumor size is decided with the pathological examination is not provided. Thereafter, the resulting tumor size may affect the accuracy of the statistical analysis when comparing these methods.

2. The tumor size with the ultrasound and FFDM is assessed only for the BC-Mp in the work. Discussions about similar work show different results with other types of breast cancer. What are the main differences between them? For example, the measurement method, tumor position, tumor morphology or other. It is best to provide their comparison results among different data when these factors are set to be the same.

3. Even if the evaluation of the tumor size with the ultrasound or the FFDM imaging method is valuable in a sense, does it affect the diagnosis and therapy method as well as the conclusion for the BC-Mp? More joint indications from more imaging modalities with multi scales need to be explored.

4. The expression can be improved. Also the structure of the manuscript can be improved. There are some confusing or unclear expressions in the manuscript. Some abbreviations, for example ‘pT’, appear multiple times with different full names.

Comments on the Quality of English Language

The expression can be improved.  There are some confusing or unclear expressions in the manuscript. Some abbreviations, for example ‘pT’, appear multiple times with different full names.

Author Response

Dear Reviewer,

Please find our reply attached,

Yours sincerely,

Miroslawa Puskulluoglu MD, PhD on behalf of the authors

Reviewer 2 Report

Comments and Suggestions for Authors

The abstract needs quantification. The table 1 may be supplemented by limitations. section 3.3 requires more enhancement with concepts. The Discussion may include limitation and further research direction with pathological concerns and clinical findings. The conclusion requires modifications in line with objectives.

Comments on the Quality of English Language

Nil

Author Response

(The authors gave the same response as above.)

Reviewer 3 Report

Comments and Suggestions for Authors

The authors present a useful study assessing the accuracy of tumour size with FFDM and USS for metaplastic breast cancer. They show low accuracy especially for large tumours. Which could result in inadequate surgery.

Specific points

1.       ‘The majority of breast malignancies originate from epithelial components [3]. Ductal carcinoma (not-otherwise specified [NOS]) accounts majority all cases, while lobular carcinoma represents 8% of cases.’ -This seems rather low for lobular. Please provide a reference

2.       Is this really the conclusion from study 15? ‘Imaging features of BC-Mp on FFDM and US were benign.’

3.       ‘Way of tumor initial detection’ -replace ‘way’ in table 2 with ‘method’

4.       Figure 4 spelling of pat(h)omorphology

5.       ‘Only the study conducted by Langlands and colleagues included a larger number of patients with BC-Mp with data from their imaging studies however, the authors there did not endeavor to address the question of which examination could more (significantly) predict tumor size as assessed through pathological examination.’ -suggest replace ‘significantly’ with accurately.

6.       Do the authors have data on BRCA1/2 testing? A recent study found a high rate of BRCA1 https://pubmed.ncbi.nlm.nih.gov/37460658/ This has implications for treatment with PARPi

7.       The authors found that all their metaplastic tumors were HER2-. Other studies in table 1 showed HER2- rates as low as 40%. Please clarify if this was due to lack of HER2 testing rather than HER2+

Comments on the Quality of English Language

Good

Author Response

(The authors gave the same response as above.)

Round 2

Reviewer 1 Report

Comments and Suggestions for Authors

The authors have revised the whole manuscript and described a point by point response to the comments. So I recommend it for publication.

Reviewer 2 Report

Comments and Suggestions for Authors

All the corrections are included and the paper may be accepted.

Comments on the Quality of English Language

Nil